# Flash Invariant Point Attention

**Andrew Liu, Axel Elaldi, Nicholas T Franklin, Nathan Russell**
**Gurinder S Atwal, Yih-En Andrew Ban, Olivia Viessmann**
Flagship Pioneering
Cambridge, MA, United States
`[anliu,aelaldi,nfranklin,nrussell,matwal,aban,oviessmann]@flagshippioneering.com`

## Abstract

Invariant Point Attention (IPA) is a key algorithm for geometry-aware modeling in structural biology, central to many protein and RNA models. However, its quadratic complexity limits the input sequence length. We introduce FlashIPA, a factorized reformulation of IPA that leverages hardware-efficient FlashAttention to achieve linear scaling in GPU memory and wall-clock time with sequence length. FlashIPA matches or exceeds standard IPA performance while substantially reducing computational costs. FlashIPA extends training to previously unattainable lengths, and we demonstrate this by re-training generative models without length restrictions and generating structures of thousands of residues. FlashIPA is available at `https://github.com/flagshippioneering/flash_ipa`.

## 1 Introduction

Invariant Point Attention, IPA, is a geometry-aware attention operation that has been the workhorse of generative structural design models for proteins and RNA, initially popularized by Alphafold2 [1] and AlphaFold-Multimer [2], and subsequently widely adopted across structural biology modeling. Amongst them are structure prediction models like OpenFold and ESMFold for proteins [3, 4], RhoFold for RNA [5], generative protein backbone models such as FrameDiff[6], FrameFlow [7, 8], the FoldFlow family [9, 10], FrameDiPT [11], Proteus [12], FADiff [13], Genie [14], IgDiff [15], GAFL [16], P2DFlow [17], RNA generative models such as RNA-FrameFlow [18], and scoring models like lociPARSE [19]. A list of models that rely on IPA for their structure modeling is provided in Appendix A.1. The advantage of IPA is its roto-translational ($SE(3)$) invariant representation of the molecular structure, enforcing the idea that rotations and translations of a molecule results in an equivalent structure prediction. This inductive geometric bias accelerates training and improves performance in limited data settings, as is the case for structurally resolved biomolecules. IPA's quadratic scaling ($O(L^2)$) limits its scalability, rapidly exhausting GPU memory when modeling longer biomolecules. As of May 2025, 42% of structures within the PDB have more than 512 residues, and 33% with more than 756 and 23% with more than 1024 (see Appendix A.2). Most trainings across the literature reduce their data to chains and cropped structures below 512 residues. Despite this, they still commonly run on costly multi-GPU setups with trainings that span from days up to a month [5, 6].

Such engineering compromises, namely truncating biologically relevant lengths, limiting dataset sizes, and relying on expensive computational infrastructure, restrict the progress of the field. In this work, we propose a recasting of the original IPA algorithm to a simple attention form to leverage off-the shelf I/O reduction methods like FlashAttention, which replace the quadratic $O(L^2)$ with an effective linear $O(L)$ scaling behavior. We show empirically that FlashIPA exceeds the validation performance of IPA in benchmarking models and datasets. We then demonstrate the memory and

compute time efficiency by retraining benchmarking models more efficiently and without length restrictions on the data.

We provide FlashIPA as an importable uv package at `https://github.com/flagshippioneering/flash_ipa` with an API similar to existing repositories using IPA to facilitate drop-in usage.

## 2 Preliminaries

Introduced by AlphaFold2 [1], IPA is a specialized attention mechanism designed to preserve 3D geometric relationships directly in transformer-based models for structural biology applications. IPA enforces invariance under 3D rotations and translations by utilizing a learnable local coordinate frame representation for each residue (point) within proteins and RNAs.

Formally, a function $f : \mathcal{X} \to \mathcal{Y}$ is *invariant* to transformations in the abstract group $G$ if $f(g \cdot x) = f(x)$ for all $g \in G$. This differs from *equivariance* to $G$, where the transformations of the group commute with $f$, or $f(g \cdot x) = g \cdot f(x)$ for all $g \in G$. Here, we are concerned with the transformations $T$ (referred to as *"frames"*) in the special Euclidean group SE(3), a continuous group of rigid transformations that includes rotations and translations, but not reflections. Intuitively, the output of an SE(3)-invariant function does not change under rotations and translations, which is desirable for biomolecular structures, that do not change function under those transformations.

### 2.1 Frame representations for proteins and RNA

Meaningful frames for biomolecules are commonly defined on the backbone structures. In proteins, each residue frame is typically determined by three of the four atoms from the backbone, namely the alpha carbon ($C_\alpha$), nitrogen ($N$), and carbon ($C$), with $C_\alpha$ at the origin. The frame transformation $T$ maps the standard positions of these backbone atoms into their actual positions in global coordinates: $[N, C, C_\alpha, O] = T \cdot [N', C', C'_\alpha, O']$. The frame has a rotation and translation component $T = (R, t)$. The rotation component of the transformation, $R$, is computed using Gram–Schmidt orthonormalization on vectors formed by $C_\alpha - N$ and $C_\alpha - C$ bonds, while the position of the oxygen ($O$) is captured by the torsion angle around the $C_\alpha - C$ bond.

For RNA, the backbone structure involves 13 atoms per nucleotide, leading to more complex flexibility. RNA-FrameFlow [18], define RNA frames using the atoms $C3'$, $C4'$, and $O4'$, identified as the least variable backbone positions. The rest of the RNA backbone geometry can then be parameterized efficiently through a set of eight torsion angles.

### 2.2 IPA algorithm

The IPA algorithm is a 1D sequence-to-sequence transformation. It transforms an input sequence representation $s_i$, local frames $T_i$, and pair representations $z_{ij}$ (with sequence indices $i, j \in 1, ..., L$) into an output sequence $\tilde{s}_i$. Each attention head, indexed by $h \in 1, ..., H$, involves linear transformations of the input sequence to produce scalar queries, keys, values ($\mathbf{q}_i^h, \mathbf{k}_i^h, \mathbf{v}_i^h \in \mathbb{R}^c$) and Euclidean (geometric) counterparts ($\vec{\mathbf{q}}_i^{hp}, \vec{\mathbf{k}}_i^{hp}, \vec{\mathbf{v}}_i^{hp} \in \mathbb{R}^3$). Additionally, the pair representation $\mathbf{z}_{ij} \in \mathbb{R}^{L \times L \times d_z}$ contributes to a bias term $b_{ij}^h$ through a linear projection:

$$\text{Single rep:} \quad \mathbf{q}_i^h, \vec{\mathbf{q}}_i^{hp}, \mathbf{k}_i^h, \vec{\mathbf{k}}_i^{hp}, \mathbf{v}_i^h, \vec{\mathbf{v}}_i^{hp} \leftarrow \text{Linear}(\mathbf{s}_i) \tag{1}$$

$$\text{Pair rep. (bias term):} \quad b_{ij}^h \leftarrow \text{Linear}(\mathbf{z}_{ij}) \tag{2}$$

$$\text{IPA update:} \quad a_{ij}^h = \text{softmax}_j \underbrace{\left( w_L \left( \frac{1}{\sqrt{c}} \mathbf{q}_i^{h\mathsf{T}} \mathbf{k}_j^h + b_{ij}^h - \frac{\gamma^h w_C}{2} \sum_p \left\| T_i \circ \vec{\mathbf{q}}_i^{hp} - T_j \circ \vec{\mathbf{k}}_j^{hp} \right\|_2^2 \right) \right)}_{O(L^2)}$$

$$\tag{3}$$

$$\text{Attention aggregation:} \quad \tilde{\mathbf{o}}_i^h = \sum_j a_{ij}^h \mathbf{z}_{ij}, \quad \mathbf{o}_i^h = \sum_j a_{ij}^h \mathbf{v}_j^h, \quad \vec{\mathbf{o}}_i^{hp} = T_i^{-1} \circ \sum_j a_{ij}^h \left( T_j \circ \vec{\mathbf{v}}_j^{hp} \right)$$

$$\tag{4}$$

Concat, project, return: $\quad \tilde{\mathbf{s}}_i = \text{Linear}\left(\text{concat}_{h,p}\left(\tilde{\mathbf{o}}_i^h, \mathbf{o}_i^h, \vec{\mathbf{o}}_i^{hp}, \left\|\vec{\mathbf{o}}_i^{hp}\right\|_2\right)\right)$

$$(5)$$

The proof of SE(3) invariance of the IPA transformation can be found in the appendix of AlphaFold2 [1]. This form of IPA incurs $O(L^2)$ I/O and memory complexity due to the explicit materialization of the quadratic attention matrix in equation 3.

## 2.3 FlashAttention and I/O reduction

The quadratic scaling of the attention operation is a well-known challenge in deep learning. Operations are generally I/O-bound on the GPU, with performance primarily limited by data transfers between the GPU's high-bandwidth memory (HBM) and its static random access memory (SRAM). I/O reduction is typically achieved through kernel fusion, which reduces memory traffic by combining multiple operations into a single CUDA kernel. FlashAttention [20, 21] achieves kernel fusion via an online and tiled computation of the softmax, which, instead of materializing the full quadratic $\mathbf{M} = \text{softmax}\left(\mathbf{q}\mathbf{k}^\mathsf{T}\right)$ matrix, performs an equivalent computation by accumulating partial contributions to the output $\mathbf{Y} = \mathbf{M}\mathbf{V}$ one tile at a time. This technique improves scalability and performance for transformer-based models. We provide the Flash-Attention pseudo-code in appendix A.3. Building on FlashAttention-1, FlashAttention-2 further reduced the number of non-matmul FLOPs, increased parallelism across thread blocks, and distributed work between warps to reduce communication through shared memory [21].

## 2.4 FlashIPA: combining geometry-awareness and efficiency

Our efficiency gains come from expressing the entire IPA update in a form that is amenable to FlashAttention (i.e. a factorized version of form $\mathbf{q}^\mathsf{T}\mathbf{k}$), which then computes the attention update without materializing any quadratic attention matrix. To do so, we rewrite the softmax argument in the IPA update (eq. (3)) as a single inner product. We also need to parameterize the pair representation (bias term eq. (2)) $\mathbf{z}_{ij} \in \mathbb{R}^{L \times L \times d_z}$ in a factorized form $\mathbf{z}_{ij} = \mathbf{z}_i^{1\mathsf{T}}\mathbf{z}_j^2$, where $\mathbf{z}_i^1, \mathbf{z}_j^2 \in \mathbb{R}^{L \times r \times d_z}$. Here, $r$ can be interpreted as the "rank" of the factorization, and is the dimension on which we perform the contraction. We then expand the sum of squared norms in the third term of eq. (3) and collect terms, resulting in an equivalent update rule, given in Algorithm 1. Note that factorizing $\mathbf{z}_{ij}$ allows not only for combining the softmax terms, but also allows us to compute the pair representation part of the attention as $\tilde{\mathbf{o}}_i^h = \sum_j a_{ij}\mathbf{z}_{ij} = \mathbf{z}_i^{1\mathsf{T}}\left(\sum_j a_{ij}\mathbf{z}_j^2\right)$, avoiding materializing any quadratic object. As a result, the attention components from eq. (1) are lifted to $\hat{\mathbf{q}}_i^h, \hat{\mathbf{k}}_j^h \in \mathbb{R}^{c+5N_{query}+rd_z}, \hat{\mathbf{v}}_i^h \in \mathbb{R}^{c+3N_{value}+rd_z}$. Regular attention is then computed on the lifted components.

With the new update in the form of regular attention, we then leverage FlashAttention to perform the computation. This leads to significant speedups in wall-clock time and since the online computation of FlashAttention avoids materializing the quadratic attention matrix, we only need to store the queries, keys, and values in the GPU's HBM, incurring only $O(L)$ memory.

## 2.5 Factorizing the pair representation

General forms of pair representations $\mathbf{z}_{ij}$ typically involve quadratic memory complexity and are not inherently decomposable. However, common implementations of IPA often compute these quadratic representations from components that intrinsically scale linearly, implying redundancy in the resulting quadratic tensor $\mathbf{z}_{ij}$ (eq. (2)). This suggests that pair representations can be efficiently approximated through low-rank factorization in latent space. In particular, common pairwise tensors in structural biology modeling, such as distance matrices and contact maps, are either low-rank or sparse due to the smooth, local nature of physical interactions like electrostatics and steric constraints. This motivates a simplified factorization strategy for initializing and updating $\mathbf{z}_{ij}$. Instead of explicitly constructing and storing a full quadratic tensor of shape $(B, L, L, d)$, we represent each feature (e.g., positional encodings, distograms, diffusion time embeddings in generative models, a.s.o.) using two lower-dimensional tensors (pseudo-factors) each of shape $(B, L, r, d)$, obtained via linear projections. We show empirically that this parametrization, despite relaxing certain inductive biases, recovers downstream performance with substantially lower memory consumption.

---
**Algorithm 1** FlashIPA with factorized pair representations
---

Given input sequence $\mathbf{s} \in \mathbb{R}^{L \times d_{in}}$, factorized pair representation $\mathbf{z}_{ij} = \mathbf{z}_i^{1\top}\mathbf{z}_j^2$ (where $\mathbf{z}^1, \mathbf{z}^2 \in \mathbb{R}^{L \times r \times d_z}$ and $\mathbf{z} \in \mathbb{R}^{L \times L \times d_z}$), frames $T_i = (R_i, \mathbf{t}_i), \forall i, j \in \{1, ..., L\}$

1: $\mathbf{q}_i^h, \vec{\mathbf{q}}_i^{hp}, \mathbf{k}_i^h, \vec{\mathbf{k}}_i^{hp}, \mathbf{v}_i^h, \vec{\mathbf{v}}_i^{hp} \leftarrow \text{Linear}(\mathbf{s}_i)$

2: $\mathbf{b}_i^{h1} \leftarrow \text{Linear}(\mathbf{z}_i^1)$

3: $\mathbf{b}_i^{h2} \leftarrow \text{Linear}(\mathbf{z}_i^2)$

4: $\hat{\mathbf{q}}_i^h \leftarrow \text{concat}\left( \mathbf{q}_i^h, \left\{ T_i \circ \vec{\mathbf{q}}_i^{hp} \right\}_{p=1}^{N_{\text{Query}}}, \left\{ \left\| T_i \circ \vec{\mathbf{q}}_i^{hp} \right\|_2^2 \right\}_{p=1}^{N_{\text{Query}}}, \mathbf{1}_{N_{\text{Query}}}, \mathbf{b}_i^{h1} \right)$

5: $\hat{\mathbf{k}}_i^h \leftarrow \text{concat}\left( \frac{w_L}{\sqrt{c}}\mathbf{k}_i^h, \left\{ \gamma^h w_L w_C T_i \circ \vec{\mathbf{k}}_i^{hp} \right\}_{p=1}^{N_{\text{Query}}}, \frac{-\gamma^h w_L w_C}{2}\mathbf{1}_{N_{\text{Query}}}, \left\{ \frac{-\gamma^h w_L w_C}{2} \left\| T_i \circ \vec{\mathbf{k}}_i^{hp} \right\|_2^2 \right\}_{p=1}^{N_{\text{Query}}}, \mathbf{b}_i^{h2} \right)$

6: $\hat{\mathbf{v}}_i^h \leftarrow \text{concat}\left( \mathbf{v}_i^h, \left\{ T_i \circ \vec{\mathbf{v}}_i^{hp} \right\}_{p=1}^{N_{\text{Values}}}, \mathbf{z}_i^2 \right)$

7: $\hat{\mathbf{o}}_i^h \leftarrow \text{FlashAttention}(\hat{\mathbf{Q}}^h, \hat{\mathbf{K}}^h, \hat{\mathbf{V}}^h)_i$

8: $\mathbf{o}_i^h, \vec{\mathbf{o}}_i^{hp}, \tilde{\mathbf{o}}_i^h \leftarrow \text{split}(\hat{\mathbf{o}}_i^h)$

9: $\vec{\mathbf{o}}_i^{hp} \leftarrow T_i^{-1}\vec{\mathbf{o}}_i^{hp}$

10: $\tilde{\mathbf{o}}_i^h \leftarrow \mathbf{z}_i^{1\top}\tilde{\mathbf{o}}_i^h$

11: $\tilde{\mathbf{s}}_i \leftarrow \text{Linear}\left( \text{concat}_{h,p}\left( \tilde{\mathbf{o}}_i^h, \mathbf{o}_i^h, \vec{\mathbf{o}}_i^{hp}, \left\| \vec{\mathbf{o}}_i^{hp} \right\|_2 \right) \right)$

12: Return $\tilde{\mathbf{s}}$

---

In many structure models $\mathbf{z}_{ij}$ is updated multiple times by a non-linear EdgeTransition module that is not part of IPA. Because of that each factor is usually highly non-linear with respect to the input, despite the low-dimensional contraction. Thus, we still retain substantial representation power with this factorization. The benefits of highly nonlinear low-dimensional factors is well motivated in the literature: Instead of large head dimensions and linear factors, i.e. $q, k = \text{Linear}(X)$, as is the case for transformers, many state space models, such as Mamba [22] use factors that are non-linear transformations of the input (e.g. $q, k = \text{Swish}(\text{Conv1d}(X))$, and much smaller contraction dimensions (often $r = 16$).

We also note that most implementations of IPA rely on pair representations $\mathbf{z}_{ij}$ that depend on distograms computed from pairwise distances. To simultaneously ensure invariance and avoid computing activations on the full distogram, we only keep the distances of the $k$ nearest neighbors $(B, L, k, n_{bins})$ instead of $(B, L, L, n_{bins})$. In practice, we compute these using a full $L \times L$ distance matrix during pre-processing, which does not impact the runtime complexity of FlashIPA. We keep track of neighbor identities by applying positional encodings on their indices. Emphasizing nearest neighbor distances is also motivated by the fact that local geometry (e.g. chemical bonds) is more important than global geometry when it comes to validity of biomolecular structures. We surmise that in future work it may be beneficial to compress all $\frac{L(L-1)}{2}$ distances into factors without materializing the pairwise matrix, in a manner similar to FlashAttention.

## 3 Experiments

### 3.1 FlashIPA is SE(3) invariant and scales linearly in memory and wall-clock time

We tested the SE(3)-invariance by applying random roto-translations to input Gaussian point clouds and measuring the output deviation after applying a single layer of the original IPA and FlashIPA. Original IPA output deviation was $< 10^{-6}$ and FlashIPA $< 10^{-3}$. We evaluate the scaling behavior in GPU memory and wall-clock time by passing single-sample batches of increasing lengths $L$ through original and FlashIPA, see Fig. 1. We use a polynomial fit (green and red dotted lines), and find an approximate GPU memory scaling for FlashIPA of $y\,[\text{MB}] = -7 \cdot 10^{-12} \cdot L^2 + 7.5 \cdot 10^{-2} \cdot L$ versus original IPA $y\,[\text{MB}] = 2.4 \times 10^{-3} \cdot L^2 + 1.4 \cdot 10^{-2} \cdot L$. The computational complexity of attention still remains $O(L^2)$ due to the softmax operation. FlashAttention reduces the memory complexity to $O(L)$. Practically I/O, rather than computation, often dominates runtime on GPU hardware, and this is reflected in the observed linear wall-clock scaling in Fig. 2A.

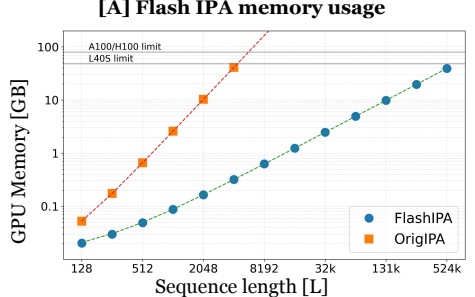
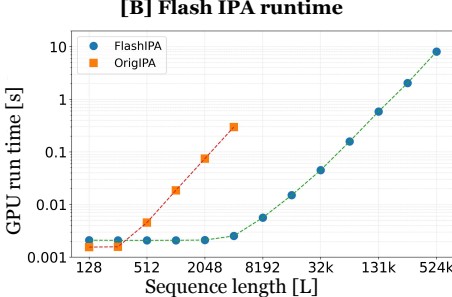

Figure 1: Scaling as a function of input sequence length on a single-sample batch forward pass. **[A]** GPU memory usage in GB. Original IPA scaled approximately quadratically with sequence length ($y\,[\text{MB}] = 2.4 \times 10^{-3} \cdot L^2 + 1.4 \cdot 10^{-2} \cdot L$), FlashIPA follows a linear trend ($y\,[\text{MB}] = -7 \cdot 10^{-12} \cdot L^2 + 7.5 \cdot 10^{-2} \cdot L$). **[B]** Wall-clock time in seconds.

## 3.2 Integration test with external repositories

We selected two recent model approaches where code and data were available for retraining of the original model and allowed for FlashIPA insertion. We chose FoldFlow [10] for proteins and RNA-FrameFlow [18] for RNAs, both are flow-matching generative backbone models. All experiments were run on L40S GPU instances with 48 GB HBM memory. FlashIPA hyperparameters were matched to the IPA parameters chosen by the original authors (embedding sizes, hidden dimensions, number of heads, etc.). For the factorization of the pair representation we tested rank $\in \{1, 2\}$, and found rank 2 and pair-wise distograms with $k = 20$ to match and partially surpass the loss convergence of original IPA.

## 3.3 FlashIPA improves performance and extends to larger proteins

We retrained the FoldFlow Base model in its original form with IPA and with FlashIPA. We reran the PDB data pre-processing pipeline by the authors, which yielded a total of 40,492 single-chain protein monomers for training. The original training used a maximum length cut-off of 512 residues, which results in a 10% reduction of the training data to 36,600 structures. We match the training strategy of the original authors and train on this reduced dataset on 4 GPUs with DDP. We kept model and train parameters identical to the published values of the authors, however for the FlashIPA version we had to make two adjustments: The original repository runs 4 blocks of IPA with a hidden dimension of 256. FlashAttention becomes incompatible with DDP at that dimension, so we reduced FlashIPA hidden dimension to 128 and used 5 blocks (instead of 4) to match model parameter sizes (17.4M versus 17.1M, theirs versus ours). To keep memory consumption at an efficient constant level, the original implementation heuristically chose an effective batch size according to the quadratic rule $\text{eff}_{bs} = \max\left[round\left(\frac{500.000 \times n_{GPUs}}{N^2}\right), 1\right]$, which kept GPU memory at approximately 90% throughout training. For FlashIPA we were able to achieve a linear effective batch size $\text{eff}_{bs} = \max\left[round\left(\frac{20.000 \times n_{GPUs}}{N}\right), 1\right]$ that resulted in similar memory consumption. For example IPA only has a batch size of 1 for a protein of length 512, whereas FlashIPA can batch together 39 samples. We found that FlashIPA converged slightly faster than IPA as expected with larger batch size, but we also compared partial loss terms, and found that particularly the number of steric clashes reduced faster for FlashIPA (we provide loss curves in the Appendix A.4). This suggests that our local k-nearest neighbour distogram emphasis is more efficient at capturing local geometry. To demonstrate the immediate enablement of linear scaling we ran one additional training with FlashIPA on the entire monomer dataset without length restrictions. The largest chain in the dataset is 8.8k residues. We follow the original authors' validation test that generates structures at varying lengths, subsequently inverse folds with Protein-MPNN [23] and forward-folds with ESMFold [4]. Generated and forward-folded structures are then aligned via the Kabsch algorithm and the resulting self-consistency RMSD (sc-RMDS) is a proxy for generation fidelity. We perform this test for all three models: original IPA and FlashIPA trained up to length 512, and FlashIPA on all data. Similar to the original paper we sampled 50 structures per length from 100 to 500, and sample 8 inverse folded sequences to forward fold per strucutre. All models are evaluated on checkpoint number 200,000. Fig. 2 A) shows that the sc-RMSD is lower for FlashIPA compared to the original IPA

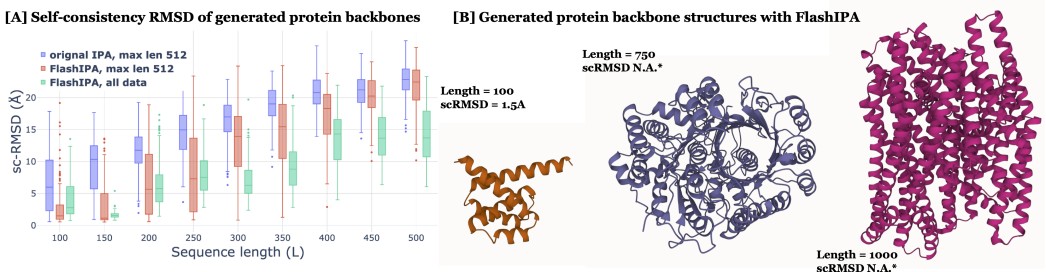

Figure 2: FoldFlow self-consistency validation after 200k optimization steps. A) The sc-RMSD of the FlashIPA (red, green) models is consistent or better than the original IPA model (blue). Extending training to larger structures with FlashIPA further improves sc-RMSD. B) Three exemplar generated backbone structures with FoldFlow FlashIPA. *ESMFold gets out of memory for lengths beyond 500+ residues and sc-RMSD could not be assessed.

trained on the same length-restricted data. Sc-RMSD is further improved when FlashIPA training is extended to the full dataset, as expected. We emphasize that FlashIPA is not designed to improve generation quality per se, but to enable training on larger and more complex structures. We suspect that the observed sc-RMSD values reflect accumulated model errors across three stages of validation: FoldFlow generation, Protein-MPNN inverse folding, and ESMFold forward folding. We could not assess sc-RMSF for larger structures, as the ESMFold also depends on IPA and ran out of memory. Overall we conclude that FlashIPA is more performant and more efficient than IPA. Fig. 2 B) shows example structures generated with the full data trained FlashIPA version.

### 3.4 FlashIPA trains more efficiently and extends to larger RNAs

We retrained the RNA-FrameFlow model using the code provided by the original authors. We ingested the same BGSU version 3.382 of the RNASolo2 dataset [24], comprising a total of 14,995 structures (see Appendix A.5). First we re-trained RNA-FrameFlow with original and FlashIPA on all structures within 40 to 150 residue (6,030 structures total), as proposed by the authors. We kept all hyperparameters consistent with the original model. We matched the authors' training setup and models were trained on 4 GPUs with DDP and an effective batch size of $4 \times 28$. 4 GPUs are necessary for effective training with IPA, but with FlashIPA it becomes feasible to run a comparable training on a single GPU instance. To demonstrate the cost efficiency we also performed an additional FlashIPA training run on the same data with a batch size of $512$ on single GPU. All models were trained for a fixed compute time of 20 hours, which resulted in approximately 156k iterations with original IPA, roughly 230k with FlashIPA on 4GPUs and 194k with FlashIPA on a single GPU. During training we found models to converge comparably with overlapping loss curves, see Appendix A.6. For an apples-to-apples comparison we validated the models using identical validity, novelty, and diversity metrics as defined in the original paper, which uses gRNAde [25] for inverse-folding of the generated structure and RhoFold [26] for forward-folding. For validity, we report the average self-consistency template modeling score (sc-TM). To do so, we generated 50 structures per length ranging from 40 to 150 residues, in increments of 10. The results are presented in Table 1 and Fig. 4. We observe comparable scores between all models. In particular, the single-GPU training matches performance at a quarter of the compute cost than the original.

To highlight the advantages of using FlashIPA in the RNA-FrameFlow model, we analyzed the GPU runtime required to generate RNA backbones based on RNA sequence length and batch size. The results are in Fig. 3. We observed improvements with FlashIPA compared to the original IPA model, especially for longer sequences and larger batch sizes. For example it takes 30 times longer to generate an RNA sequence of length 2048 with the original IPA than with FlashIPA.

We also re-trained RNA-FrameFlow with FlashIPA on the full RNASolo dataset without maximum length restrictions (the longest structures being 4,417 nucleotides) and filtering out structures shorter than $40$ nucleotides. We trained the model on 4 GPUs for 48 hours, with batch size of $4 \times 28$, resulting in approximately 310k iterations. In Fig. 4, we present samples of large generated RNA structures with lengths of 2000 and 4000 nucleotides. In comparison, the original RNA-FrameFlow can not be

| Model | Validity (↑) | Diversity (↑) | Novelty (↓) | Checkpoint | Cost |
|---|---|---|---|---|---|
| | (scTM ± std) | (qTM cluster) | (pdbTM ± std) | (#steps) | (# GPU hours) |
| RNA-FrameFlow | $0.42 \pm 0.21$ | 0.15 | $0.81 \pm 0.10$ | 156k | 80 |
| . + FlashIPA (ours) | $0.38 \pm 0.20$ | 0.14 | $0.82 \pm 0.10$ | 230k | 80 |
| . + FlashIPA single GPU ** (ours) | $0.41 \pm 0.21$ | 0.08 | $0.77 \pm 0.09$ | 194k | 20 |
| . + FlashIPA ** + All data (ours) | $0.36 \pm 0.14$ | 0.14 | $0.74 \pm 0.08$ | 310k | 192 |

Table 1: Average validity, diversity, and novelty scores for 600 generated RNAs of length $\leq 150$ with RNA-FrameFlow, with and without FlashIPA. FlashIPA provides competitive results at a fraction of the cost of an original IPA layer. ** indicates that batch size has been increased to match GPU memory capacities.

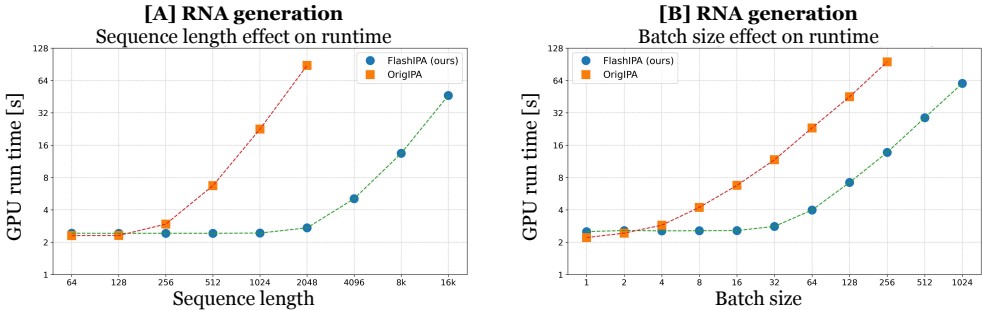

Figure 3: Scaling of FlashIPA versus IPA for RNA generation using RNA-FrameFlow model, with a number of diffusion timestep $N_T = 50$. **[A]** Impact of the generated sequence length on the generation runtime, using a batch size of 1. **[B]** Impact of the generated batch size on the generation runtime, for generated sequence of length 128.

trained or generate such long sequences. The inverse-forward-folding consistency test cannot be run for structures of thousands of residues as the validation models (gRNAde [25] for inverse-folding and RhoFold [26] for forward-folding) run out of memory at such lengths. However, we qualitatively observe that the generated structures resemble RNA, we don't necessary suspect those to be valid, as the amount of training data at such length is still comparably small. We note that sc-TM scores vary across sequence lengths, with some fluctuations (e.g., at 120 residues). This pattern is consistent with the original RNA-FrameFlow paper, which attributes it to a length imbalance in the training distribution and biases introduced by the RhoFold structure predictor used during evaluation (see [18] Appendix A.3).

## 4 Discussion

FlashIPA provides memory and wall-clock efficient $SE(3)$ invariant training on biomolecules and allows for training on structures of thousands of residues. This approach opens up the possibility of multi-chain complex modeling in situations constrained by the original IPA module. Our work permit usage of IPA in contexts where it may have been ruled out due to its scaling limitations, including representations beyond polymeric backbone that include more atomic detail.

Efficiency and scalability to long contexts has often been a neglected aspect of geometric deep learning research and many modules have quadratic to cubic complexity. An example is triangular attention used in popular models like AF-3[27], Chai-1[28] and Boltz-1 [29]. Recently, TriFast [30] was released, which uses fused triangular attention kernels to reduce I/O complexity from cubic to quadratic. We assume that a factorized version might even achieve a linear scaling. Most efficient, sub-quadratic models tend not to incorporate geometric inductive biases. Our work can be seen as a first step at bridging this divide, enabling models that respect invariances, are cost-effective, and can scale to larger or more fine-grained biomolecular systems.

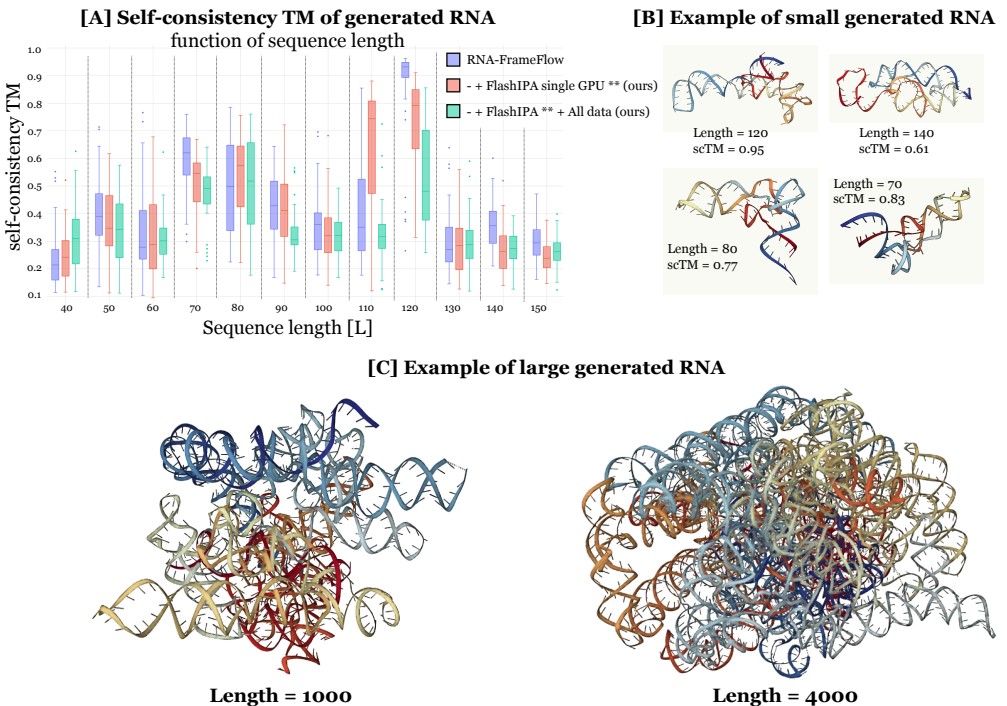

Figure 4: RNA-FrameFlow generated RNAs results. **[A]** Comparison of the scTM score depending on the generated RNA sequence length and the IPA module used. The RNA-FrameFlow and RNA-FrameFlow + FlashIPA models are both trained on only short sequences $\leq 150$, while the All data model has been trained without maximum sequence lenghth limit. We don't observe a significant difference between the different trained RNA-FrameFlow models. **[B]** Example of short generated RNA structures with our FlashIPA RNA-FrameFlow model trained on short sequences. **[C]** Example of long generated RNA structured with our FlashIPA RNA-FrameFlow model trained on the full dataset.

## 4.1 Limitations

FlashIPA relies on factorization of the pair representation, which ultimately is an approximation. Here we did not find a decrease in performance, but this may not be guaranteed for more general applications to other forms of more dense pair representations. Exploring alternative schemes specialized for certain pair representations (e.g. cross-concatenations, distograms, pair index differences) may further improve downstream performance.

Despite substantial memory and I/O savings, current FlashAttention implementations (e.g., FlashAttention2) impose a maximum head dimension of 256, due to the kernel's internal design, which expects stacked query/key/value vectors. Since our method relies on augmenting the attention components, this means that we must have $\max(c + 5N_{query} + rd_z, c + 3N_{value} + rd_z) \leq 256$. Here, we achieved competitive results using head dimensions around 200, so this was not a practical limitation. However, the Triton implementation of FlashAttention2 based on AMD CDNA (MI200, MI300) and RDNA GPUs allows for extending the head dimension to arbitrary size. We expect these improvements to apply more broadly to a wider class of GPU architectures in the months to come. Unlocking larger head dimension would allow us to increase our factorization rank, thereby better approximating dense pair representations and closing potential gaps against quadratic IPA.

Furthermore, we note that despite achieving $O(L)$ in I/O and memory, the underlying compute cost is still $O(L^2)$ due to the softmax. Removing the softmax and using linear attention variants such as Mamba would allow us to achieve $O(L)$ cost in both compute and memory, and is thus a promising avenue of future work.

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

# A Appendix

## A.1 List of biomolecular design models with IPA structure modules

Table 2: A list of models that are based on IPA structure modeling. Models were identified via GitHub advanced search, searching for commonalities in code implementations.

| | Model name | Modeling approach | Training data | Train length cut-offs |
|---|---|---|---|---|
| Protein | OpenFold [3] | protein structure prediction | 130k single protein chains (PDB) | <256, fine-tuning <384,<512,<5120 |
| | FrameDiff [6] | diffusion for backbone generation | 20k monomers (PDB) | 60-512 |
| | FrameDiPT [11] | diffusion for backbone inpainting | 32k monomers (PDB) | 60-512 |
| | Proteus [12] | diffusion for backbone generation | 51k single chains from oligo- and monomers (PDB) | 60-512 |
| | FADiff [13] | diffusion for multi-motif backbone generation | Identical to FrameDiff | 60-512 |
| | Genie [14] | diffusion for backbone generation | 26k monomers (PDB) | 60-512 |
| | IgDiff [15] | FrameDiff finetuning for antibody backbone generation | Synthetic antibody structures from folding | Not reported |
| | FrameFlow [8, 7] | flow-matching for backbone generation and motif scaffolding | Identical to FrameDiff | 60-512 |
| | FoldFlow [10, 9] | Stochastic flow matching for backbone generation | 22.2k monomers (PDB) | 60-512 |
| | GAFL [16] | Geometric algebra flow matching for backbone generation | Identical to FrameDiff | 60-512 |
| | Multiflow [31] | Discrete and continuous flow matching for joint sequence-backbone generation | 18.7k monomers (PDB) | 60-384 |
| | P2DFlow [17] | flow matching for backbone ensemble generation | 100 MD simulation trajectories from ATLAS | Not reported |
| RNA | Rho-Fold [5] | RNA structure prediction | 5.5k chains (PDB) | <1024 |
| | RNA-FrameFlow [18] | flow matching for backbone generation | 5.3k original structures + 1.1k cropped structures (PDB) | 40-150 |
| | lociPARSE [19] | Locality-aware IPA for structure scoring | 52k synthetic structures from 1.4k sequence targets | <200 |

## A.2 Residue counts of macromolecules in the PDB

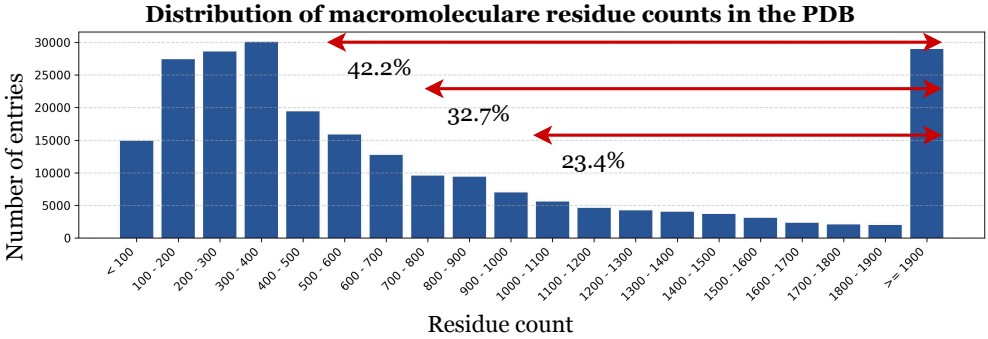

Figure 5: Distribution of protein residue counts of the proteins resolved in the PDB (Figure reproduced and adapted from the data at RCSB PDB Statistics: Sequence length distribution; https://www.rcsb.org/stats/distribution-residue-count; accessed 6 May 2025).

## A.3 FlashAttention Algorithm

Pseudo-code for the FlashAttention kernel, an online tiled computation of the softmax, which, instead of materializing the full quadratic $\mathbf{M} = \text{softmax}\left(\mathbf{q}\mathbf{k}^{\mathsf{T}}\right)$ matrix, performs an equivalent computation by accumulating partial contributions to the output $\mathbf{Y} = \mathbf{M}\mathbf{V}$ one tile at a time [20, 21]. A good introduction with line-by-line explanations can also be found here [32].

**Algorithm 2** Flash Attention-2 via online softmax (Dao 2024)

Given $\mathbf{Q}, \mathbf{K}, \mathbf{V} \in \mathbb{R}^{L \times d}$, $T_c, T_r$ row blocks of size $B_r$, $T_c$ column blocks of size $B_c$.

1: **for** $i = 1$ to $T_r$ **do**
2:      Load $\mathbf{Q}_i$ from HBM to SRAM
3:      $\mathbf{O}_0 = \mathbf{0}_{B_r \times d}, l_0 = \mathbf{0}_{B_r}, m_0 = -\infty_{B_r}$
4:      **for** $j = 1$ to $T_c$ **do**
5:          Load $\mathbf{K}_j, \mathbf{V}_j$ from HBM to SRAM.
6:          On chip, compute $\mathbf{S}_i^{(j)} = \mathbf{Q}_i \mathbf{K}_j^T$
7:          On chip, compute $m_i^{(j)} = \max\left(m_i^{(j-1)}, \text{rowmax}\left(\mathbf{S}_i^{(j)}\right)\right)$, $\tilde{\mathbf{P}}_i^{(j)} = \exp\left(\mathbf{S}_i^{(j)} - m_i^{(j)}\right)$,
     $l_i^{(j)} = e^{m_i^{(j-1)} - m_i^{(j)}} l_i^{(j-1)} + \text{rowsum}\left(\tilde{\mathbf{P}}_i^{(j)}\right)$
8:          On chip, compute $\mathbf{O}_i^{(j)} = \text{diag}\left(e^{m_i^{(j-1)} - m_i^{(j)}}\right)^{-1} \mathbf{O}_i^{(j-1)} + \tilde{\mathbf{P}}_i^{(j)} \mathbf{V}_j$
9:      **end for**
10:      On chip, compute $\mathbf{O}_i = \text{diag}\left(l_i^{(T_c)}\right)^{-1} \mathbf{O}_i^{(T_c)}$
11:      Write $\mathbf{O}_i$ to HBM as $i$th block of $\mathbf{O}$
12: **end for**
13: Return $\mathbf{O}$

## A.4    FoldFlow training convergence

We provide loss curves for the training of the FoldFlow base model with original and FlashIPA below. FlashIPA converged slightly faster for the same number of optimization steps, which is expected from the bigger effective batch size, but we also found in particular local loss terms, such as the steric clash loss to decrease noticably more efficient.

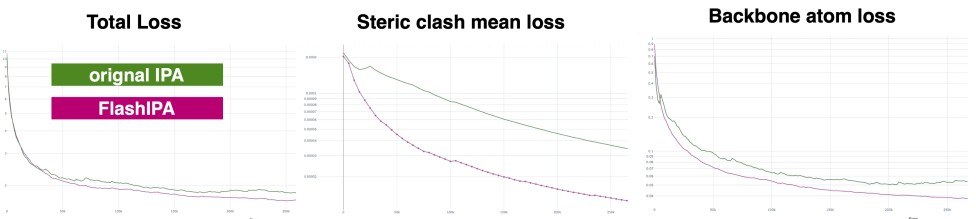

Figure 6: Loss behaviour for FoldFlow model training.

## A.5    Nucleotide residue counts of RNA structures in the RNASolo2 dataset.

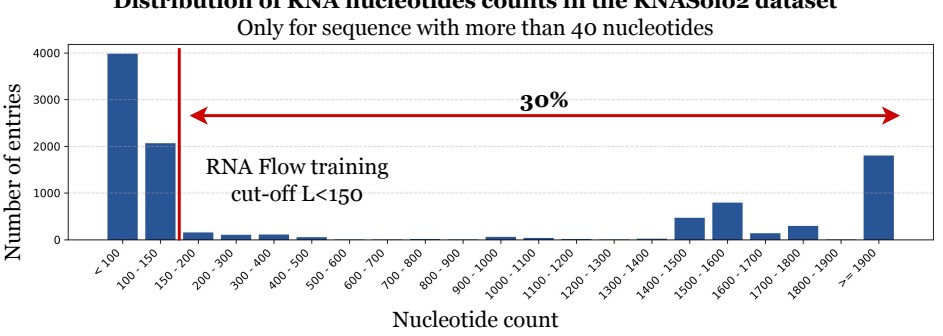

Figure 7: Distribution of nucleotide residue counts of the RNA in the RNASolo2 dataset, filtering out the short structures of length $< 40$ nucleotides. The training cut-off of RNA Flow discards all sequences of length $> 150$, accounting for $30\%$ of the dataset.

### A.6 RNA-FrameFlow training convergence

We provide loss curves for the training of the RNA-FrameFlow base model with original and FlashIPA below. At similar hyperparameters and hardware, RNA-FrameFLow with and without FlashIPA behaves similarly.

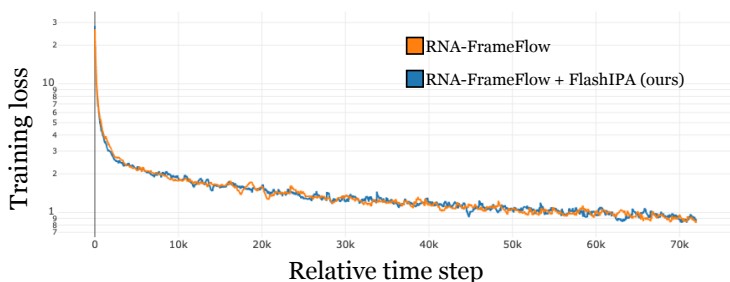

Figure 8: Loss behaviour for RNA-FrameFlow model training.

