# OpenReview forum: "Flash Invariant Point Attention"
_NeurIPS.cc/2025/Conference — NeurIPS 2025 spotlight_

### Official Review · Reviewer_N3zQ · 2025-06-27

**Clarity:** 2
**Significance:** 3
**Originality:** 3
**Rating:** 5
**Confidence:** 5

**Summary:**

This paper presents FlashIPA, a computationally efficient reformulation of Invariant Point Attention (IPA), which has been widely adopted in protein and RNA structure modeling. FlashIPA replaces the original quadratic attention mechanism with a FlashAttention-compatible variant that uses low-rank factorization of the pairwise attention bias term z_{ij}. This enables SE(3)-invariant attention to scale linearly in memory with sequence length, allowing modeling of long biomolecular sequences previously limited by IPA’s memory footprint.

The authors demonstrate that FlashIPA preserves geometric invariance, matches or exceeds the performance of IPA in protein and RNA generative models, and enables training and inference on sequences with thousands of residues.

**Questions:**

1. Did you try collapsing the pair bias term into the q/k vectors directly and training with a single unified attention score? How does that compare in performance?
2. Are there examples or tasks (e.g., complex assemblies or RNA tertiary structures) where evolving the pairwise state would be essential? Could your framework support that in future work?
3. Could you provide more intuition or theoretical grounding for why the factorized z^1_i{}^\top z^2_j is a good approximation of the original pair tensor z_{ij}? Under what assumptions does this hold?

**Ethical Concerns:**

["NO or VERY MINOR ethics concerns only"]

**Final Justification:**

In light of the authors’ clarifications, I have revised my score to an acceptance.

**Limitations:**

Yes!

**Paper Formatting Concerns:**

Nothing so far!

**Quality:**

3

**Strengths And Weaknesses:**

STRENGTHS
1. The memory and compute limitations of IPA are well-known, and the need to scale structure models to longer sequences is urgent.
2. The key technical insight (recasting IPA as a form of dot-product attention compatible with FlashAttention) is sound and well-executed.
3. The paper includes thorough experiments on FoldFlow and RNA-FrameFlow, showing comparable or improved performance at a fraction of the compute cost.
4. Providing a plug-and-play implementation and benchmarking setup improves reproducibility and impact.

WEAKNESSES
1. In practice, all attention terms are collapsed into a single residue embedding s_i, and the pair embedding z_{ij} is not retained across layers. The “modularity” argument for keeping pairwise and nodewise paths separate appears more architectural than essential, given that the final representation is merged anyway.
2. It would be useful to evaluate a version of the model where the pairwise bias (z¹/z²) is simply folded into the q/k projection. This would test whether the factorized form offers any real gain over just using a single expressive attention term.
3. Only ranks r = 1, 2 are tested. It’s unclear whether larger ranks would provide meaningful gains, or if performance saturates early. A deeper ablation could clarify this.
4. While the paper seems to pose FlashIPA as a drop-in replacement for standard IPA, this only holds for models that do not maintain an explicit z_{ij} representation. In models like AlphaFold, where z_{ij} is evolved through the Evoformer via TriangleAttention and other operations, FlashIPA’s assumption of factorized z¹/z² vectors is not directly compatible. The paper does not address how to approximate or decompose an existing z_{ij}, nor does it evaluate the impact of such an approximation on downstream structure quality. This undermines the generalizability of the method and should be considered a notable practical limitation.

---

> ### Author Rebuttal · Authors · 2025-07-29
>
> ### Weaknesses/Questions
> 1. ***Only ranks $r=1,2$ are tested. It's unclear whether larger ranks would provide meaningful gains.***
> Lower ranks $(r = 1, 2)$ have lower memory usage, and as wHigher ranks start to reduce the memory efficiency gains. We were able to recover original IPA convergence with $r=2$ (see Appendix A4 and A6), we do not expect further computational gains with higher rank approximations. Nonetheless, we will add look into running a full $rank = 3$ training so we can add it to the convergence plots in the appendix for completeness.
>
> 2. ***FlashIPA does not apply to models that do not maintain an explicit $z_{ij}$ representation. In models like AlphaFold, where $z_{ij}$ is evolved through the Evoformer via TriangleAttention and other operations, FlashIPA’s assumption of factorized $z^1/z^2$ vectors is not compatible. The paper does not address how to approximate or decompose an existing $z_{ij}$, nor does it evaluate the impact of such an approximation on downstream structure quality. This undermines the generalizability of the method and should be considered a notable practical limitation.***
> We agree with the reviewer — these questions are of particular interest and are the subject of our ongoing research. Alphafold switched from using IPA in the AF-2 Evoformer to using Triangular Attention and Multiplication in AF-3. TriangleAttention and the TriangularMultiplication algorithm are distinct algorithms to IPA and they require a different set of kernel fusion approaches, that cannot be covered by a FlashAttention drop-in. Thus, we consider triangular attention out of scope for this paper, but hope to provide an answer to this question in a future paper.
> However, we disagree that this undermines the generalizability of FlashIPA, because IPA and triangular attention and multiplication are two very different modeling approaches ranging from their invariance properties to the explicit algorithm. We are optimistic that lessons from FlashIPA will be useful improving the efficiency of Triangle attention.
>
> 3. ***Did you try collapsing the pair bias term into the q/k vectors directly and training with a single unified attention score?***
> It's an interesting comment, we didn’t test this. We do expect it to perform similarly since it follows the same intuition of leveraging the compressibility of the pair representation.
>
> 4. ***Are there examples or tasks (e.g., complex assemblies or RNA tertiary structures) where evolving the pairwise state would be essential? Will your framework support this?***
> That's a good question. In structural biology most interactions can be effectively modeled with sparse or low-rank pairwise representations. In large macromolecular complexes or self-assembling systems where long-range interactions in sequence space can be spatially proximal. Examples could be viral capsids, ribosomes, chromosomal assemblies, and RNA tertiary structures with complex motifs like pseudoknots. For these phenomena we'd still expect the distance/contact maps to be reasonably sparse, but that is a guesstimate and truly needs to be tested. Outside structural biology, many problems, eg in social or communication networks may require dense pairwise modeling as well, though they typically lack $SE(3)$ symmetry and fall outside the scope of our method.
>
> 5. ***Provide more intuition or theoretical grounding for why the factorized $z^1_i{}^\top z^2_j$ is a good approximation of the original pair tensor $z_{ij}$? Under what assumptions does this hold?***
> The factorized form $\mathbf{z}_{ij} \approx \mathbf{z}^1_i{}^\top \mathbf{z}^2_j$ is a good approximation when the original pairwise tensor is compressible, i.e., sparse, low-rank, or redundant due to being derived from underlying 1D sequences. More specifically, this factorized form is the optimal rank-k linear approximation (Eckart-Young-Mirsky theorem) and underlies a number of common techniques in scientific settings. In machine learning, a similar approximation underlies effective techniques such as LORA and matrix factorization in recommender systems.
> Protein and RNA structures are governed by local and smoothly varying interacting forces (sterics and electrostatics) that are intrinsically low dimensional and make the pairwise representations highly compressible and motivate the use of low-rank approximations.
> We amended the text to clarify:
> *”This suggests that pair representations can be efficiently approximated through low-rank factorization in latent space. In particular, common pairwise tensors in structural biology modeling, such as distance matrices and contact maps, are either low-rank or sparse due to the smooth, local nature of physical interactions like electrostatics and steric constraints. This motivates a simplified factorization strategy… “*

---

> > ### Comment · Reviewer_N3zQ · 2025-08-02
> >
> > Thank you for the detailed response. While many concerns were addressed, I would still appreciate more clarification on the choice to retain separate pairwise and residue paths given that all attention is ultimately collapsed into a single embedding.

---

> > > ### Author Response · Authors · 2025-08-04
> > > **Clarification why pairwise and sequence representations are kept separate and not collapsed**
> > >
> > > **Since in the end we simply have a 1D sequence $s_i$ -- why didn't we just merge in the $z_{ij}$ in the first attention layer and drop the $z_{ij}$ path later on as opposed to evolving both paths?**
> > >
> > > Every IPA layer has its own attention map ($q_i \cdot k_j + b_{ij}$, where the bias term $b_{ij}$ is a projection of $z_{ij}$). If only the $z_{ij}$ is used in the first layer and not in the rest of the layers, then the necessary biases are not passed into the subsequent layers. In this case, it is true that the final $s_i$ will contain information from $z_{ij}$, but the sequence mixing for the later layers won't be correct (since it won't be informed by prior knowledge encoded in the pair representation).
> > > Hence, $z_{ij}$ is maintained to keep the bias $b_{ij}$ that informs the sequence mixing. Evolving the $z_{ij}$ through multiple MLPs is to make sure this pair representation is expressive enough. This is in line with the original AlphaFold/OpenFold implementations. We hope this answers the reviewer's question.

---

> > > > ### Comment · Reviewer_N3zQ · 2025-08-05
> > > >
> > > > Thanks for the clarification.

---

### Official Review · Reviewer_wkTi · 2025-06-28

**Clarity:** 2
**Significance:** 4
**Originality:** 2
**Rating:** 5
**Confidence:** 3

**Summary:**

Invariant point attention (IPA) is a variant of the attention mechanism that is used in SOTA protein and RNA structure prediction methods, allowing to apply the mechanism to 3D conformations. Unfortunately, its complexity is $O(L^2)$. The authors adapt an existing variant of attention, FlashAttention, which is $O(L)$, to IPA, resulting in a method called Flash IPA (FIPA) that exhibits good performance and reduced computational cost, at the expense of introducing some approximations and apparantly a complexity that is still $O(L^2)$.

**Questions:**

It would benefit the article from adding one additional drop-in test case for both RNA and proteins. If this is not possible, why not?

What are the current practical implications of the $O(L^2)$ complexity due to the softmax?

How significant is the 256 head dimension limit in practice?

Page 4, line 129: What are these “inductive biases” that are being relaxed? Could you clarify the effect of the low-rank factorization?

Page 5, line 164: Rank 2 is quite low. Could you elaborate on why this works well?

Page 4: Could you elaborate on the use of the kNN algorithm? What algorithm was used, what is its complexity, and what are the practical and theoretical  implications?

**Ethical Concerns:**

["NO or VERY MINOR ethics concerns only"]

**Final Justification:**

The authors provided satisfactory answers to all my concerns, notably about the complexity of the final method.

**Limitations:**

Limitations are addressed in the final section (but see my first item under “Weaknesses”).

**Paper Formatting Concerns:**

The authors only use 8 pages. Therefore, I suggest to move A3 to the main paper.

**Quality:**

3

**Strengths And Weaknesses:**

Strengths

IPA is an important algorithm in SOTA 3D protein and RNA structure prediction. The authors address a an important computational bottleneck that hinders training on large macromolecules, and demonstrate its efficacy using drop-in modified versions of FoldFlow (for proteins) and RNA-FrameFlow (for RNA).

Weaknesses

There is some confusion wrt the final complexity of the method. In the final section, “Limitations”, the authors write “the the underlying compute cost is still $O(L^2)$ due to the softmax”. This should be clearly reflected in the abstract and introduction, as these now give the impression of a linear complexity.

The authors test FIPA on only one protein and one RNA prediction method.

---

> ### Author Rebuttal · Authors · 2025-07-29
>
> ### Questions
> 1. ***Can you add additional drop-in test cases for both RNA and proteins?***
> Our original plan was to test FlashIPA with OpenFold, but we ran into several issues with the repository and were unable to run training (see response to Reviewer Hu1L). We thoroughly evaluated nearly every model listed in Appendix Table 2 for retraining feasibility, which required not only complete training code and configuration files but also access to the exact training and validation data. FoldFlow and RNA-FrameFlow were the models we found to be fully reproducible end-to-end (we note that FoldFlow2 was not reproducible in our hands due to inconsistent code). If more repositories provided fully reproducible setups—including original datasets and exact training configs—we would have included a third benchmark. We’ve made FlashIPA publicly available and have already seen strong community interest and early integration efforts, so we’re optimistic that additional validations will follow from the community.
>
> 2. ***There is some confusion wrt the final complexity of the method. What are the practical implications of the $O(L^2)$ complexity due to the softmax? Isn't the model's true complexity still $O(L^2)$***
> Great point, it's well taken—we agree that the distinction between **computational** and **I/O** complexity could have been made more clear in the text. There are different notions of complexity: compute, I/O, and memory. Due to the softmax, the **computational** complexity of attention remains fundamentally **$O(L^2)$**. However, FlashAttention reduces the **I/O and memory** complexity from $O(L^2)$ to **$O(L)$** through kernel fusion. I/O is usually the dominant bottleneck in practice, and leads to effective wall-clock time improvements even though computation remains quadratic.
> We’ve reviewed the manuscript to clarify this distinction. The abstract already correctly states that FlashIPA achieves “linear scaling in GPU memory and wall-clock time,". This is supported both experimentally in Figure 2 and explicitly discussed in Section 3.1.
> We clarify these points and amended this paragraph:
> *“We use a polynomial fit (green and red dotted lines), and find an approximate GPU memory scaling for FlashIPA of $y [MB] = −7 · 10^{−12} · L^2 + 7.5 · 10^{−2} · L $ versus original IPA $y [MB] = 2.4 × 10^{−3} · L^2 + 1.4 · 10^{−2} · L$. The computational complexity of attention still remains $O(L^2)$ due to the softmax operation, but FlashAttention reduces the memory complexity to $O(L)$. Practically I/O, rather than computation, often dominates runtime on GPU hardware, and this is reflected in the observed linear wall-clock scaling in Fig. 2A”*
> By contrast, linear attention variants (which eliminate the softmax) can achieve true $O(L)$ complexity across compute, memory, and I/O by exploiting matrix multiplication associativity. Extending these ideas to support IPA-style geometric invariance would be a promising direction for future work.
>
> 3. ***How significant is the 256 head dimension limit in practice?***
> In our experiments, we obtained competitive results with a head dimension of ~200, so the 256 limit was not a practical constraint. This limitation arose only because we used the existing FlashAttention-2 kernel as-is, which required stacking multiple vectors. In principle, this stacking isn’t essential—linear I/O and memory complexity can still be achieved as long as the pair representation is factorized. A custom kernel that avoids stacking could remove this limit entirely, and we may explore that in future. We clarify this point in the manuscript and added:
> *”Despite substantial memory and I/O savings, current FlashAttention implementations (e.g., FlashAttention2) impose a maximum head dimension of 256, due to the kernel's internal design, which expects stacked query/key/value vectors. Since our method relies on augmenting the attention components, this means that we must have $\max(c+5N_{query}+rd_z, c+3N_{value}+rd_z)\leq 256$.  Here, we achieved competitive results using head dimensions around 200, so this was not a practical limitation.”*
>
> 4. ***Page 4, line 129: What are these “inductive biases” that are being relaxed? Could you clarify the effect of the low-rank factorization?***
> We call the factorization of the pair representation a relaxation of inductive bias because the original 2D matrix corresponds to highly interpretable pairwise interactions. Representing them as compressed factors loses this interpretability.
> We aren’t exactly directly factorizing the 2D matrix, in the sense that the two 1D sequences, when contracted, aren’t intended to numerically approximate the original 2D matrix. Instead, we are representing/compressing the 2D matrix using two low-dimensional 1D sequences. Such compression is justified when there is substantial redundancy or sparsity in the original 2D matrix, which is often the case.
>
> 5. ***Page 5, line 164: Rank 2 is quite low. Why does this work well?***
> For a tensor of size $(B,L,r,d)$, you can think of the “effective inner dimension” as really being $r\times d$. Thus, even with small $r$, the effective inner dimension can still be quite large. We really only introduced the new $r$ dimension so that we can contract the tensors once and still be left with another $d$ dimension to work with.
>
> 6. ***Page 4: Could you elaborate on the use of the kNN algorithm? What algorithm was used, what is its complexity, and what are the practical and theoretical implications?***
> Great question. In our implementation we used the simplest form of kNN where we actually materialized the $L\times L$ distance matrix on-the-fly. This is done only once as a preprocessing step and is thus considered separate from the FlashIPA algorithm complexity. Also, if you precomputed these knn indices and stored them in the dataset you would have a memory complexity of $O(L\cdot k)$.
> You could also compute the $k$ nearest indices in a block-wise streaming fashion. This should be very easy to add (we might just add this in the code).
> Alternatively, you can also try to use approximate kNN methods, such as those that use hashing (LSH) or FAISS, which will achieve on-the-fly memory complexity of $O(L\cdot k)$. But again, since the k-nearest neighbors can be precomputed beforehand to achieve $O(L\cdot k)$ memory, the exact method used for computing them isn’t very relevant to our algorithm.
> We amended the manuscript to clarify this point:
> *”…, we only keep the distances of the $k$ nearest neighbors $(B,L,k,n_{bins})$ instead of $(B,L,L,n_{bins})$. In practice, we compute these using a full $L \times L$ distance matrix during pre-processing, which does not impact the runtime complexity of FlashIPA.”*
>
> ### Paper Formatting Concers
> 1. ***You only use 8 pages, move A3 (FlashAttention background) to the main text***
> Fair point. We moved A3 FlashAttention explanations up into the main text, we left the pseudo-code for the online tiled computation of the softmax in the appendix as it not our original work and not strictly necessary for the reader to fully grasp.

---

> > ### Comment · Reviewer_wkTi · 2025-08-01
> > **Thank you**
> >
> > I am satisfied with the answers and maintain my rating.

---

### Official Review · Reviewer_Hu1L · 2025-06-30

**Clarity:** 3
**Significance:** 3
**Originality:** 3
**Rating:** 5
**Confidence:** 3

**Summary:**

This paper introduces FlashIPA, a novel reformulation of Invariant Point Attention (IPA) designed to overcome IPA's quadratic computational complexity, which limits its application to long protein and RNA sequences. FlashIPA achieves linear scaling in GPU memory and wall-clock time with sequence length by factorizing the IPA algorithm and leveraging hardware-efficient FlashAttention. The proposed method aims to make geometry-aware modeling more accessible and scalable for structural biology. Experimental results demonstrate that FlashIPA matches or surpasses the performance of standard IPA, significantly reduces computational costs, and enables training and generation of biomolecular structures with thousands of residues, a feat previously difficult to achieve due to memory constraints. The authors provide FlashIPA as a readily usable package.

**Questions:**

NA

**Ethical Concerns:**

["NO or VERY MINOR ethics concerns only"]

**Final Justification:**

I appreciate auhtors' responses and raise my score accordingly.

**Paper Formatting Concerns:**

The font used in this paper appears to deviate from the official NeurIPS templates.

**Quality:**

3

**Strengths And Weaknesses:**

### Strength

- Integrating FlashAttention into IPA is the biggest contribution. The core strength of FlashIPA is its ability to reduce the computational complexity of IPA from quadratic (O(L2)) to effectively linear (O(L)) in terms of GPU memory and wall-clock time with sequence length. This is a major breakthrough for handling long biomolecular sequences.
- FlashIPA not only achieves significant efficiency gains but also matches or even exceeds the performance of standard IPA in Protein/RNA de novo benchmark.
- The linear scaling of FlashIPA allows training and generation of structures with thousands of residues, extending to lengths previously unattainable due to memory limitations of standard IPA. It provides integration tests with established protein (FoldFlow) and RNA (RNA-FrameFlow) generative models.

### Weakness

This paper exhibits no major weaknesses. However, a significant omission is the lack of testing FlashIPA within AlphaFold2, which is the most prevalent and advanced platform for long-protein modeling. While *de novo* design is important, it typically does not necessitate the generation of long protein/RNA structures, and the resulting designs predominantly feature alpha-helical conformations.

---

> ### Author Rebuttal · Authors · 2025-07-29
>
> ### Summary
> Thank you for the positive feedback. Besides the AF2 retraining, given the comment ***“this paper has no major weaknesses”*** and your enthusiasm for the significant cost reduction and readily usable package — could you clarify what improvements would raise this review from “borderline accept” to “accept” or “strong accept”?
>
> ### Weaknesses
> 1. ***A significant omission is the lack of testing FlashIPA within AlphaFold2, the most prevalent platform for long-protein modeling***
> We completely agree—evaluating FlashIPA within AF2 would have been ideal. Unfortunately, the AF2 training code is not publicly available. We found that the repo for OpenFold, an open source replication of AF2, has several problems we were unable to resolve, including: outdated PyTorch Lightning dependencies, import errors, and difficulties around template generation. Despite opening issues on the OF GitHub, we were unable to resolve them. For apples-to-apples comparison on loss convergence behaviour we would have also needed to retrain AF2 for one to two epochs at least. In practice this requires massive compute resources (≥128 GPUs), which further makes such experiments infeasible with the original IPA implementation. Our hope is that FlashIPA will make it possible for more researchers to train these models on affordable hardware in the future.
>
> ### Paper Formatting Concerns
> 1.  ***The font deviates from the NeurIPS templates***
> Thanks, we used an outdated template and have updated it, which fixed the font :)

---

### Official Review · Reviewer_szby · 2025-07-02

**Clarity:** 2
**Significance:** 3
**Originality:** 3
**Rating:** 5
**Confidence:** 3

**Summary:**

This paper introduces flash invariant point attention (FlashIPA), which reformulates IPA to leverate FlashAttention and thus achieve linear scaling in GPU memory wrt sequence length (the sequences being AA sequences here, for instance). The paper is well-written, with a clear structure and addresses a clear need in the structural biology x AI community, and the package is also well-structured and documented. I am not able to well-review the software engineering aspects of the work but I can at least comment on its significance which is high.

Edit: I have updated my score following the thoughtful response and review from the authors.

**Questions:**

- Are the reference/generated structures aligned for computing the sc-RMSD (e.g, experiments in Fig. 3, Fig. 5)?
- What is the reason for the big variance in sc-RMSD (e.g., outliers), even for the same sequence length? Is it specific structural motifs (or flexible protein regions) that make some systems very inconsistent? And are the "outliers" for the original IPA and FlashIPA experiments the same (that is, is it the same structures that are very inconsistent and/or reliably consistent)?
- Related to the above point, it is also unclear to me if for each sequence length, does this indicate just a single structure with that length (but many times), or are there many structures with a given length that were evaluated?

**Ethical Concerns:**

["NO or VERY MINOR ethics concerns only"]

**Final Justification:**

I have updated my score following the thoughtful response and review from the authors from a 4 to a 5.

**Limitations:**

See above, some things are unclear in the paper.

**Quality:**

3

**Strengths And Weaknesses:**

Major Points
- The "anonymous" linked repo was not, in fact, anonymous, but contained a bibtex for the arxiv version of this work.
- Is it not alarming that none of the generated protein backbones are very self-consistent (i.e., sc-RMSD>4 Å), even with FlashIPA? I guess it is not clear to me why this task was chosen as the baseline for FlashIPA, because we do not have a fully accurate ground truth, but I guess it can be thought of as a lower bound? Some clarification in the paper would be good.
- In Table 1, it would be good to indicate the cost along with the metrics, because the metrics just show worse performance without emphasizing that with FlashIPA you got those results more quickly (but, how much more quickly?).
- Is TM score here the same as template modeling score? I would clarify this in an article for this venue.
- It is not clear to me why in Figure 5 there is such a big difference between the scTM scores at some lengths (e.g., 120). Is not what you expect a similar scTM score across all methods (just knowing that it was obtained more quickly with scTM)? So what is the reason for the discrepancy?
- Again, it is unclear to me in the RNA experiments whether the number of RNAs generated "per length" correspond to the same sequence or different sequences.

Minor Points
- Inconsistent cross-referencing to figures/tables in text (sometimes "Figure" sometimes "Fig.")

---

> ### Author Rebuttal · Authors · 2025-07-29
>
> ### Major points
> 1. The linked repo contained a bibtex for an arxiv version.
> Apologies, we removed the bibtex information in the linked repo.
> 2. ***a) Is it not alarming that none of the generated protein backbones are very self-consistent (i.e., sc-RMSD>4 Å), even with FlashIPA?***
> FlashIPA does indeed produce self-consistent structures with sc-RMSDs in the range of 1.5–2.5 Å for lengths 100–150 (see Fig. 3A), well below the 4 Å threshold mentioned. But the reviewer is correct, this level does not hold throughout all length ranges, especially once lengths increase (less training samples).
> ***b) Why was this task chosen?***
> This task was chosen to match the benchmark used by FoldFlow, which is a widely adopted protein backbone generation model. Our goal was to ensure a direct, apples-to-apples comparison. We have amended the manuscript to clarify this with the following *“For an apples-to-apples comparison we validated the models using identical validity, novelty, and diversity metrics as defined in the original paper, which uses gRNAde [25] for inverse-folding of the generated structure and RhoFold [26] for forward-folding.”*
> ***c) There is no ground truth for generation, is sc-RMSD in Figure 3A an indicator of accuracy lower bound?***
> FlashIPA is not a generative model itself, but an efficiency framework that enables scaling up training—both in data volume and structure length. This scaling translates into improved sc-RMSD performance when applied within the FoldFlow → protein-MPNN → ESMFold validation pipeline: from ~6–22 Å with the original IPA training set to 1.5–13 Å with the full dataset enabled by FlashIPA (see Fig. 3A, purple vs. green box plots). We believe the observed performance reflects accumulated errors/variability across the three evaluation model stages rather than a lower bound on achievable accuracy.
> ***d) ”Some clarification in the paper would be good.”***
> We take the point for more clarity and added this paragraph:
> *"We emphasize that FlashIPA is not designed to improve generation quality per se, but to enable training on larger and more complex structures. We suspect that the observed sc-RMSD values reflect accumulated model errors across three stages of validation: FoldFlow generation, Protein-MPNN inverse folding, and ESMFold forward folding.”*
>
> 3. ***In Table 1, it would be good to indicate the cost along with the metrics***
> Very useful feedback, we added two columns to Table 1 indicating the training step at evaluation and the number of GPU training hours (cost). Now it is clear how similar results are obtained at a quarter of the cost.
>
> 4. ***Clarify TM-score***
> We added details to the manuscript
>
> 5. ***What is the reason for the scTM discrepancy across models in Figure 5?***
>  RNA-FramFlow is a sequence-agnostic generative model and samples a backbone for a given length. We follow the evaluation protocol of the original paper, which samples 50 structures per length. Additional variability in generated structures across models is expected because the models were not trained with matched optimization steps, but under fixed 20-hour wall-clock budgets to reflect equivalent computational cost. As a result, FlashIPA models underwent more training iterations (156k for original IPA vs. 230k and 194k for FlashIPA variants), likely affecting generalization behavior.
> The original RNA-FrameFlow paper also reported length variability in TM scores due to length distribution imbalances in the training set and length-dependent biases in the RhoFold structure predictor:
> *”We believe the over representation of certain lengths in the training distribution causes the fluctuation of TM-scores. We can also partially attribute this to the inherent length bias of RhoFold; see Appendix A.3. With better structure predictors, we expect more samples to be valid.”
> ***b) what explains the discrepancy across models at some lengths, e.g. 120?***
> Our model trained with FlashIPA and all data (Figure 5A, green box) sees a more diverse training distribution than the original model that had to exclude data above 150 residues. We assume that the FlashIPA model has reduced overfitting effects seen at certain lengths (e.g., 120 residues).
> To clarify and avoid confusion we added this paragraph:
> *”We note that sc-TM scores vary across sequence lengths, with some fluctuations (e.g., at 120 residues). This pattern is consistent with the original RNA-FrameFlow paper, which attributes it to a length imbalance in the training distribution and biases introduced by the RhoFold structure predictor used during evaluation (see [18] Appendix A.3).”*
>
> 6. ***It is unclear in the RNA experiments whether the number of RNAs generated per length correspond to the same sequence or different sequences***
> The generated RNAs per length do not correspond to the same sequence. RNA-FrameFlow is a generative, sequence-agnostic model that samples diverse backbone structures conditioned only on length. For evaluation, sequences are inferred post hoc via gRNAde (inverse folding), and structures are then forward-folded with RhoFold, described in the manuscript in section 3.4.
>
> ### Minor points
> 1. ***sometimes you use Figures and sometimes Fig.***
> Thanks, fixed.
>
> ### Questions
> 1. ***Are the reference/generated structures aligned for computing the sc-RMSD?***
> Yes, we added this detail to the manuscript:
> ”Generated and forward-folded structures are then aligned via the Kabsch algorithm…”
>
> 2. ***What is the reason for the variance in sc-RMSD even for the same sequence length? Are the outliers for the original IPA and FlashIPA the same?***
> Please see our answer to the Major Points, #5b. We have not explicitly investigated whether outlier structures are consistent across IPA and FlashIPA, but this would be an interesting avenue for future analysis. However, we’d suspect that this will likely be the case, as it is highly likely an artifact of the training data distribution and not the algorithmic approach to the method.
>
> 3. ***It is also unclear to me if for each sequence length, does this indicate just a single structure with that length (but many times)?***
> Please see our previous answers to Major Points #6. Both FoldFlow and RNAFrameFlow are generative, sequence-agnostic methods. For each target sequence length, we generate 50 distinct backbone structures, each sampled independenly.

---

### Decision · Program_Chairs · 2025-09-17

**Decision:**

Accept (spotlight)

**Comment:**

The submission presents FlashIPA, a factorized reformulation of Invariant Point Attention (IPA) that leverages hardware-efficient FlashAttention to achieve linear scaling in GPU memory and wall-clock time with sequence length. The IPA layers offer invariance under 3D rotations and translations and have been widely used in AlphaFold2 and many other protein neural networks. The proposed FlashAttention allows training models on longer proteins and RNAs more efficiently. The reviewers have unanimously voted for the acceptance of the paper. Given the significance of the contributions, I am happy to support the recommendation.